# Analysis of the protein composition of the spindle pole body during sporulation in *Ashbya gossypii*

**Dario Wabner**[�], **Tom Overhageböck**[�], **Doris Nordmann, Julia Kronenberg, Florian Kramer, Hans-Peter Schmitz**[ORCID]*

Department of Genetics, University of Osnabrück, Osnabrück, Germany

او These authors contributed equally to this work.

* hans-peter.schmitz@biologie.uni-osnabrueck.de

**Data Availability Statement:** All relevant data are within the manuscript and its Supporting Information files.

## Abstract

The spores of fungi come in a wide variety of forms and sizes, highly adapted to the route of dispersal and to survival under specific environmental conditions. The ascomycete *Ashbya gossypii* produces needle shaped spores with a length of 30 μm and a diameter of 1 μm. Formation of these spores relies on actin and actin regulatory proteins and is, therefore, distinct from the minor role that actin plays for spore formation in *Saccharomyces cerevisiae*. Using *in vivo* FRET-measurements of proteins labeled with fluorescent proteins, we investigate how the formin AgBnr2, a protein that promotes actin polymerization, integrates into the structure of the spindle pole body during sporulation. We also investigate the role of the *A. gossypii* homologs to the *S. cerevisiae* meiotic outer plaque proteins Spo74, Mpc54 and Ady4 for sporulation in *A. gossypii*. We found highest FRET of AgBnr2 with AgSpo74. Further experiments indicated that AgSpo74 is a main factor for targeting AgBnr2 to the spindle pole body. In agreement with these results, the *Agspo74* deletion mutant produces no detectable spores, whereas deletion of *Agmpc54* only has an effect on spore length and deletion of *Agady4* has no detectable sporulation phenotype. Based on this study and in relation to previous results we suggest a model where AgBnr2 resides within an analogous structure to the meiotic outer plaque of *S. cerevisiae*. There it promotes formation of actin cables important for shaping the needle shaped spore structure.

## Introduction

Fungi use spores as a specialized cellular form for survival and dissemination. Often these spores are highly adapted to a certain habitat or a specific route of distribution, resulting in distinct spore shapes and sizes. As a consequence, a wide variety of fungal spore morphologies exist. How fungal spores are formed is studied in detail using *Saccharomyces cerevisiae* as a model organism. In *S. cerevisiae*, sporulation is coupled to meiosis of diploid cells. Sporulation is initiated under nitrogen starvation in combination with a poor carbon source and results in the typical tetrad consisting of four round spores. A key structure in the process of the

**Funding:** This work was supported by grant SCHM 2388/2-1 from the Deutsche Forschungsgemeinschaft (DFG) and the Open Access Publishing Fund of Osnabrück University. The funders had no role in study design, data collection and analysis, decision to publish, or preparation of the manuscript.

**Competing interests:** The authors have declared that no competing interests exist.

formation of these spores, and an essential component for spore morphology, is the spindle pole body [1]. The fungal spindle pole body is the equivalent to a centrosome and the microtubule organization center during vegetative growth in yeast. However, during sporulation and with the onset of meiosis II the spindle pole body is reconstructed into an organizing center for membrane formation. This reconstruction results in loss of the Gamma-tubulin small complex, which is achieved by the removal of Spc72, the Gamma-tubulin small complex receptor [2]. Spo21, Mpc54, Spo74 and Ady4 then form a novel layer at the spindle pole body, the meiosis II outer plaque (MOP;[2–5]). The MOP functions as a vesicle docking complex, which is used for docking secretory vesicles which fuse to form an initial prospore membrane. The prospore membrane grows by fusing increasingly more vesicles until the complete spore structure is formed. This process involves many components of the secretory machinery of the cell [4, 6–10]. Recent work in *S. cerevisiae* has helped substantially towards understanding these processes that lead to fungal spore formation. However, since *S. cerevisiae* spores are of a rather simple round shape, they do not help to explain how more complex spore morphologies are formed. We have started to investigate this question, utilizing the filamentous ascomycete *Ashbya gossypii* as a model [11, 12].

The spores of *A. gossypii* have a distinct needle shape with a rather large average length of about 30 μm and a diameter of about 1 μm at the widest part [12]. This spore shape is probably an adaptation for the transmission of the fungus by sucking insects [13]. By analyzing several mutants with distinct defects in spore formation, we were able to identify several factors like the formin AgBni1, the small GTPases AgRho1a and AgRho1b and the paxillin homolog AgPxl1, as important factors for spore morphology [12]. Since all the above proteins are involved in regulation of the actin cytoskeleton, it would appear that actin is more important during sporulation when the form of the spore becomes more complex. This is also supported by the fact that we found another formin protein, AgBnr2, as a factor which localizes to the spindle pole body with the onset of sporulation. Here, the protein is required for formation of actin cables, which seem to be important for the process of shaping the spore's form. This suggests that, during sporulation, the spindle pole bodies in *A. gossypii* are reconstructed from a microtubule- to an actin organization center. Using two-hybrid studies, we identified several potential interaction partners of AgBnr2 at the spindle pole body [11]. In the present study our objective is to further characterize the interaction of AgBnr2 with spindle pole body components by *in vivo* Förster-Resonance-Energy-Transfer (FRET) measurements. As described above, in *S. cerevisiae* the spindle pole body is the starting point for formation of the MOP and the prospore membrane. We have demonstrated in previous experiments that AgSpo21, a component of the MOP, is essential for sporulation and shows interaction with AgBnr2 in 2-hybrid experiments [11]. Therefore, we included a selection of potential components of this structure into our analysis, focusing on potential adaptations that might lead to the typical *A. gossypii* spore shape. Looking for homologs to MOP components from *S. cerevisiae* in *A. gossypii* reveals that homologs to *SPO74*, *MPC54*, *ADY4 and SPO21* all exist in *A. gossypii*. However, the protein sequences all differ in length and the maximum identity is only 20%, which is too low to even speculate about similarities in protein function [11]. Nevertheless, RNA-seq data under sporulation condition reveals that all four MOP-components in *A. gossypii* are induced significantly, but even under induced conditions, some of them are not found within the highly expressed proteins. *AgSPO74*, for example, only ranks within the lowest third of all expressed proteins under these conditions [14].

In the following we will use the term Sporulation Outer Plaque (SOP) for a potentially existing homologous MOP-structure, until more proof of the existence of meiosis in *A. gossypii* is presented. This is due to the fact that it is not clear if meiosis, or a similar process, is linked to sporulation in *A. gossypii*, which is commonly thought to be haploid [14]. Nevertheless, recent

work suggests that nuclei within the *A. gossypii* mycelium can display a large degree of hetero-geneity and aneuploidy, which seems to be reversible by sporulation, suggesting that at least a process similar to meiosis may occur before or during sporulation in *A. gossypii* [15].

## Methods

### Strains and culture conditions

The construction of all *Ashbya gossypii* strains was performed using PCR-based gene targeting, as described by Wendland *et al.* [16]. For the generation of targeting cassettes for deletions or gene fusions, we used template vectors from the pAGT series [17] or pGEN3 (Wendland et al., 2000) or derivatives therefrom. The transformation of *A. gossypii* with plasmids was performed according to the protocol by Wright *et al.*[18]. All strains used for this study are listed in Table 1.

Strains were named using the following scheme: "C" and "R" before or after the protein name indicate amino- or carboxy-terminal fusions to yoClover or yomRuby2 [19], respectively. The organism specific prefix "Ag" before gene or protein names was omitted in Figures and strain names due to space limitations.

*A. gossypii* was cultured in Ashbya Full Medium (AFM; [16]) supplemented with or without 200 μg/ml geneticin (Sigma-Aldrich, Taufkirchen, Germany) or 100 μg/ml ClonNAT (Werner

**Table 1. Strain and genotypes used in this study.**

| Name | Genotype |
| --- | --- |
| *Δady4* | *Agleu2Δ; Agthr4Δ; Agady4::GEN3* |
| *Δmpc54* | *Agleu2Δ; Agthr4Δ; Agmpc54::GEN3* |
| *Δspo74* | *Agleu2Δ; Agthr4Δ; Agspo74::GEN3* |
| *Δspo74*-Bnr2R-Hhf1-GFP | *Agleu2Δ; Agthr4Δ; Agspo74::GEN3; [HIS3P-AgBNR2-yomRUBY2; HHF1-NLS-GFP; LEU2]* |
| Ady4R | *Agleu2Δ; Agthr4Δ; LEU2-HIS3P-AgADY4-yomRUBY2-GEN3* |
| Ady4RBnr2C | *Agleu2Δ; Agthr4Δ; LEU2-HIS3P-AgADY4-yomRUBY2-GEN3; [HIS3P-AgBNR2-yoCLOVER::ClonNAT; LEU2]* |
| Bnr2C | *Agleu2Δ; Agthr4Δ; LEU2-HIS3P-AgBNR2-yoCLOVER::ClonNAT* |
| BNR2G | *Agleu2Δ; Agthr4Δ; [PScHIS3-AgBNR2-GFP GEN3]* |
| Bnr2R | *Agleu2Δ; Agthr4Δ; LEU2-HIS3P-AgBNR2-yomRUBY2::ClonNAT* |
| Bnr2RBnr2C | *Agleu2Δ; Agthr4Δ; LEU2-HIS3P-AgBNR2-yoCLOVER::ClonNAT* |
| Bnr2RC | *ΔAgleu2; ΔAgthr4, [HIS3P-AgBNR2-yoCLOVER-yomRUBY2::ClonNAT; LEU2]* |
| Bnr2R-Hhf1-GFP | *ΔAgleu2; ΔAgthr4; [HIS3P-AgBNR2-yomRUBY2; HHF1-NLS-GFP; LEU2]* |
| CBnr2Bnr2R | *LEU2::HIS3P-AgBNR2-yomRUBY2::ClonNAT; [HIS3P-yoCLOVER-AgBNR2::GEN3; LEU2]* |
| CBnr2R | *ΔAgleu2; ΔAgthr4 [HIS3P-yoCLOVER-AgBNR2-yomRUBY2::GEN3; LEU2]* |
| Cmd1R | *ΔAgleu2; ΔAgthr4; [AgCMD1-yomRUBY2::GEN3; LEU2]* |
| Cmd1RBnr2C | *Agleu2Δ; Agthr4Δ; LEU2-HIS3P-AgBNR2-yoCLOVER::ClonNAT [AgCMD1-yomRUBY2::GEN3; LEU2]* |
| Cnm67R | *Agleu2Δ; Agthr4Δ; AgCNM67-yomRUBY2::GEN3* |
| Cnm67T | *Agleu2Δ; Agthr4Δ; [CNM67-tdtomato]* |
| Cnm67TBnr2G | *Agleu2Δ; Agthr4Δ; [PScHIS3-AgBNR2-GFP GEN3] [CNM67-tdtomato]* |
| Cnm67RBnr2C | *Agleu2Δ; Agthr4Δ; AgCNM67-yomRUBY2::GEN3; [HIS3P-AgBNR2-yoCLOVER::ClonNAT; LEU2]* |
| Spc72R | *Agleu2Δ; Agthr4Δ; AgSPC72-yomRUBY2::GEN3* |
| Spc72RBnr2C | *Agleu2Δ; Agthr4Δ; AgSPC72-yomRUBY2::GEN3; [HIS3P-AgBNR2-yoCLOVER::ClonNAT; LEU2]* |
| Spo74R | *Agleu2Δ; Agthr4Δ; LEU2::HIS3P-AgSPO74-yomRUBY2::GEN3* |
| Spo74RBnr2C | *Agleu2Δ; Agthr4Δ; LEU2::HIS3P-AgSPO74-yomRUBY2::GEN3 [HIS3P-AgBNR2-yoCLOVER::ClonNAT; LEU2]* |
| Tub4R | *Agleu2Δ; Agthr4Δ; AgTUB4-yomRUBY2::GEN3* |
| Tub4RBnr2C | *Agleu2Δ; Agthr4Δ; AgTUB4-yomRUBY2::GEN3; [HIS3P-AgBNR2-yoCLOVER::ClonNAT; LEU2]* |

Bioagents, Jena, Germany) at 30˚C. For use of the auxotrophic marker *leu2*, cells were cultured in synthetic minimal medium (ASC; [16]).

For FRET measurements 50 μl of a spore suspension were grown overnight in 5 ml AFM supplemented with ClonNAT or G418, if necessary. The following morning cells were washed twice with 1 ml ASC. 500 μl timelapse-medium (25% medium, 0.5% agarose, 2% Glucose) were spread on a slide. After solidification, cells were spread on the slide in 7 μl drops. The slides were incubated for 3–4 hours at 30˚C in a petri dish on a moist paper towel to avoid drying.

## DNA manipulation, plasmids and constructs

The DNA was manipulated in accordance with the methods of Sambrook et al. [20] using *Escherichia coli* DH5α as a host strain. For PCR, either the Dream Taq Polymerase or the Phusion High-Fidelity DNA Polymerase (Thermo Scientific, Schwerte, Germany) was used. Oligonucleotides were synthesized by Microsynth (Balgach, Switzerland). Plasmid DNA was isolated from *E. coli* using the GeneJET Plasmid Miniprep Kit (Thermo Scientific, Schwerte, Germany). DNA sequencing was performed by GATC-Biotech (Cologne, Germany). Sequences, plasmids, primers and details of plasmid constructions are available upon request.

## Growths assays

A tip of a toothpick of mycelial material was placed in the middle of an agar plate containing either AFM or ASC dependent on the strain. The plates were placed on a LED light table in a closed box equipped with a digital-camera in a 30˚C room. Images were taken every 3 hours. Each batch of 6 plates contained at least two wild-type strains as a reference and a lineal for length calibration. The increase in mycelial surface area was determined by image analysis in FIJI (ImageJ 2.0.0-rc-65/1.51s; [21]). Statistical calculations were performed using R version 3.5.0 [22, 23] in combination with Rstudio desktop version 1.1.453 [24], with modules ggplot2, plyr, and dplyr [25, 26].

## Staining of *A. gossypii* spores

All staining procedures were performed as described in detail previously [12].

## Microscopy

The setup used for fluorescence microscopy consisted of a Zeiss Axioplan 2 (Carl Zeiss, Jena, Germany) equipped with a 100× alpha-Plan Fluor objective (NA 1.45), a motorized Bioprecision 2 stage controlled by a MAC5000 controller (Ludl Electronic Products, Ltd., Hawthorne, USA), differential-interference contrast and filters suitable for the detection of yoClover and yomRuby2. Light sources were a pE-100wht (CoolLED Ltd, Andover, Great Britain) for DIC and a Spectra light engine (Lumencor, Beaverton, USA) for excitation of fluorescence. The signals were recorded by an ORCA-Flash4.0 LT PLUS Digital CMOS camera from Hamamatsu Photonics (Herrsching am Ammersee, Germany). The setup was controlled by MetaMorph software (v.7.7.6.0, Molecular Devices, Sunnyvale, CA, USA). Pictures were taken for each strain in the order: brightfield, yomRuby2, FRET and yoClover.

## Determination of FRETR

Initial trials using the software FRETSCAL for automated determination of FRETR [27] were not successful because FRETSCAL was optimized for the dimensions of *S. cerevisiae* spindle pole bodies and the nuclei within an *A. gosspyii* mycelium are very mobile, leading to

movement of the SPB-signal between acquisition of the green, red or FRET channels. This required adaptation of the FRETSCAL algorithm to *Ashbya gossypii* as follows: We omitted the step of automatic selection of spindle pole bodies and selected spindle pole bodies manually by cropping regions of 200 x 200 pixels from the images. These images were then subjected to image analysis using FIJI (ImageJ 2.0.0-rc-65/1.51s; [21]) similar to FRETSCAL. Briefly this involved background substraction, identification of the image maximum and a Gaussian fit. The intensity values derived from this image analysis were then loaded into R version 3.5.0 [22, 23] in combination with Rstudio desktop version 1.1.453 [24], with modules ggplot2, plyr, and dplyr [25, 26] for Statistical calculations and graphics. FRETR values were calculated as described in detail by Muller et al. [28], where FRETR is the quotient of the signal in the FRET-channel divided by the total spillover. The total spillover is calculated by the spillover measured in the green channel multiplied with the signal of the green channel added to the spillover measured in the red channel multiplied with the signal of the red channel. The spill-over values for green and red channels have been determined before with strains carrying only the fusion to Clover or to mRuby2. They represent the amount of signal that is visible in the FRET channel in these strains.

## Results

Based on previous 2-hybrid studies [11], we selected a variety of proteins that have shown interaction with either the full length- or an aminoterminal fragment (1–270) of AgBnr2 for investigation of *in vivo* FRET analysis. At the onset of our studies the best available FRET-pair was mRuby2 combined with Clover [29]. Therefore, we decided to fuse the Clover-gene to AgBnr2 and the mRuby2-gene to the genes of the following proteins: AgTub4, AgSpc97 and AgSpc98 from the γ-Tubulin complex, the outer plaque component AgSpc72 and AgCnm67, a component of the inner layer 1. To test a possible interaction with the SOP, we added AgSpo74, AgMpc54 and AgAdy4 to the selection. Because strain construction is more time consuming in *A. gossypii* than in *S. cerevisiae*, and because existing strains cannot be crossed, we limited strain construction to carboxy-terminal fusions except for the investigation of AgBnr2 homo-dimerization. For the same technical reasons, only the mRuby2-fusions were performed directly on the chromosome and the labeled AgBnr2 was added later as an episomal copy.

As negative control from the central plaque we labeled AgCmd1 with mRuby2 at the amino-terminal end also using an episomal copy of the gene. For a positive control mRuby2 and Clover were simultaneously fused to the carboxy-terminal end of AgBnr2. To characterize the self-interaction of AgBnr2 that we observed in previous 2-hybrid studies, we also constructed a version of the gene carrying mRuby2 at the carboxy-terminal end, a version carrying mRuby2 at the amino-terminal end and a version combining carboxy-terminal and amino-terminal fusions of Clover and mRuby2 to a single protein. For strains showing no or only a weak fluorescence signal, the constitutive *HIS3*-promotor from *S. cerevisiae* was fused in front of the gene to increase signal strength. Even this did not lead to a detectable signal for AgSpc97, AgSpc98 and AgMpc54. We also wanted to include AgSpo21, for which we had shown 2-hybrid interaction with AgBnr2 in a previous work [11] as a positive control. However, we were unable to obtain a signal for AgSpo21 when fused to mRuby2, which prevented it from inclusion into our analysis. Table 2 lists all genes tested for inclusion in this study. Genes that were labled successfully and included in the study are marked. The Table also includes relative mRuby2-signal intensities.

All strains carrying a single label were first tested for growth defects and for visibility of the fluorescence signal under the microscope. Growth of most strains did not differ significantly

**Table 2. Genes investigated for this study.**

| Gene | Fusionprotein | (a) amino- /(c) carboxy-terminal fusion | Promoter | Relative mRuby2 signal intensity | Signal/ included in analysis |
|------|---------------|-----------------------------------------|----------|----------------------------------|------------------------------|
| AgADY4 | mRuby2 | c | ScHIS3 | 1 | + |
| AgBNR2 | mRuby2 | c | ScHIS3 | 11.3 | + |
| AgBNR2 | Clover | a | ScHIS3 | - | + |
| AgBNR2 | Clover | c | ScHIS3 | - | + |
| AgBNR2 | mRuby2-Clover | c | ScHIS3 | - | + |
| AgBNR2 | yEGFP | c | ScHIS3 | - | + |
| AgBNR2 | mRuby2, Clover | a, c | ScHIS3 | - | + |
| AgCNM67 | tdtomato | c | AgCNM67 | 1.9 | + |
| AgCNM67 | mRuby2 | c | AgCNM67 | - | - |
| AgCMD1 | mRuby2 | c | AgCMD1 | 3.7 | + |
| AgMPC54 | mRuby2 | c | ScHIS3 | - | - |
| AgSPC42 | mRuby2 | c | ScHIS3 | - | - |
| AgSPC97 | mRuby2 | c | ScHIS3 | - | - |
| AgSPC98 | mRuby2 | c | ScHIS3 | - | - |
| AgSPO21 | mRuby2 | c | ScHIS3 | - | - |
| AgSPO74 | mRuby2 | c | ScHIS3 | 6.3 | + |
| AgTUB4 | mRuby2 | c | AgTUB4 | 3.6 | + |

from a wildtype-reference (S1 Fig). Only the fusion to AgCnm67 showed slower growth, especially in older mycelia. Strains displaying a signal were then transformed with a plasmid carrying AgBnr2 labeled with Clover and again tested for growth (S2 Fig). Again, the strain with the labeled AgCnm67 grew significantly worse than the other strains, but there seemed to be no additional negative effect by the labeled AgBnr2. Because this strain also did not yield measurable FRET, in contrast to our previous FRET-measurements utilizing tdtomato fused to AgCnm67 and GFP to AgBnr2 as fusion proteins, we also used the latter strains for measurements of FRET between AgCnm67 and AgBnr2.

Fig 1 shows examples of the Clover, Ruby and FRET-images (respectively tdtomato, GFP and FRET for AgCnm67 and AgBnr2) of all strains displaying usable signals for the final FRET-measurements. The images shown in Fig 1 are optimized for visibility of the signal and do not reflect signal strength as measured in the FRET-experiments.

These strains were then subjected to FRET measurements as described in the Materials and Methods section. As can be seen in Fig 2 there is high variability of the FRETR values we observed. Two factors may have contributed to this observation. One possible factor was that we had to rely on expression from a plasmid for some of the constructs, resulting in different copy numbers and thus different expression levels of the proteins. The second factor is influenced by the fact that interaction of Bnr2 with the SPB occurs primarily during sporulation and we had no possibility of ascertaining whether a mycelium underwent sporulation or not. Therefore, the lower values of each combination might represent the status under non-sporulation conditions, whereas higher values might come from mycelia that underwent sporulation at the time we performed our measurements.

The FRETR-values we obtained can be roughly categorized into three classes. The FRETR of AgTub4 and AgAdy4is in the same range or even below our negative control AgCmd1, which is a component of the central plaque of the SPB. FRETR values slightly higher than the negative control are seen for carboxyterminal fusions of yomRuby2 and yoClover to two different AgBnr2 variants, for AgSpc72 and for amino- and carboxyterminal fusions of yomRuby2 and yoClover to the same AgBnr2 molecule. While the median of these measurements

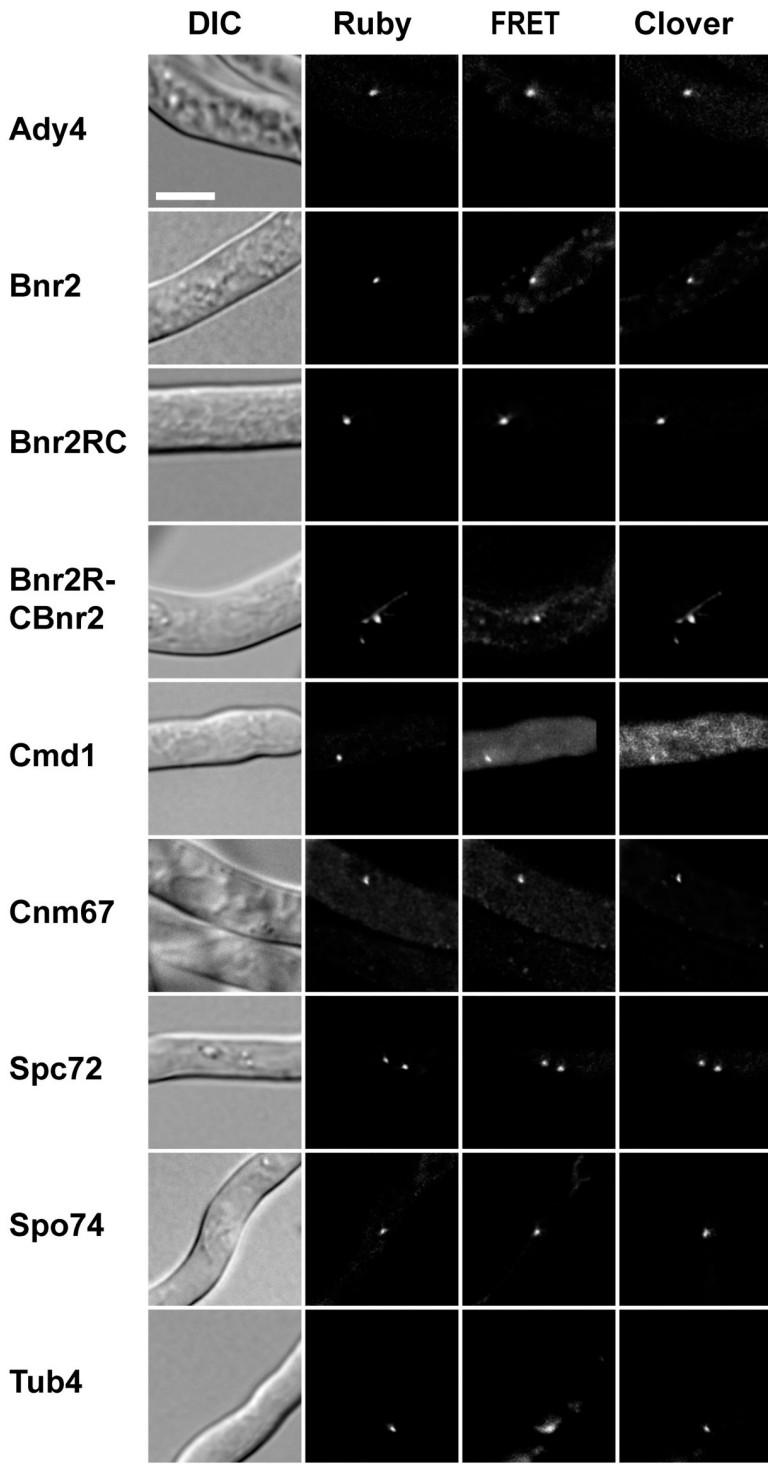

**Fig 1. Sample images of each strain used for FRET-measurements.** The indicated proteins of each strain are labeled at the carboxy-terminus with yomRuby2 and carry an additional episomal copy of AgBnr2 labeled with yoClover also at the carboxy-terminal end, except for AgBnr2RC and AgBnr2RCBnr2 that carry both labels on a single protein. Note that the intensity of the fluorescent signals is scaled for visibility and does not reflect the intensity measured, especially in the FRET channel. The scale bar represents 10 μm.

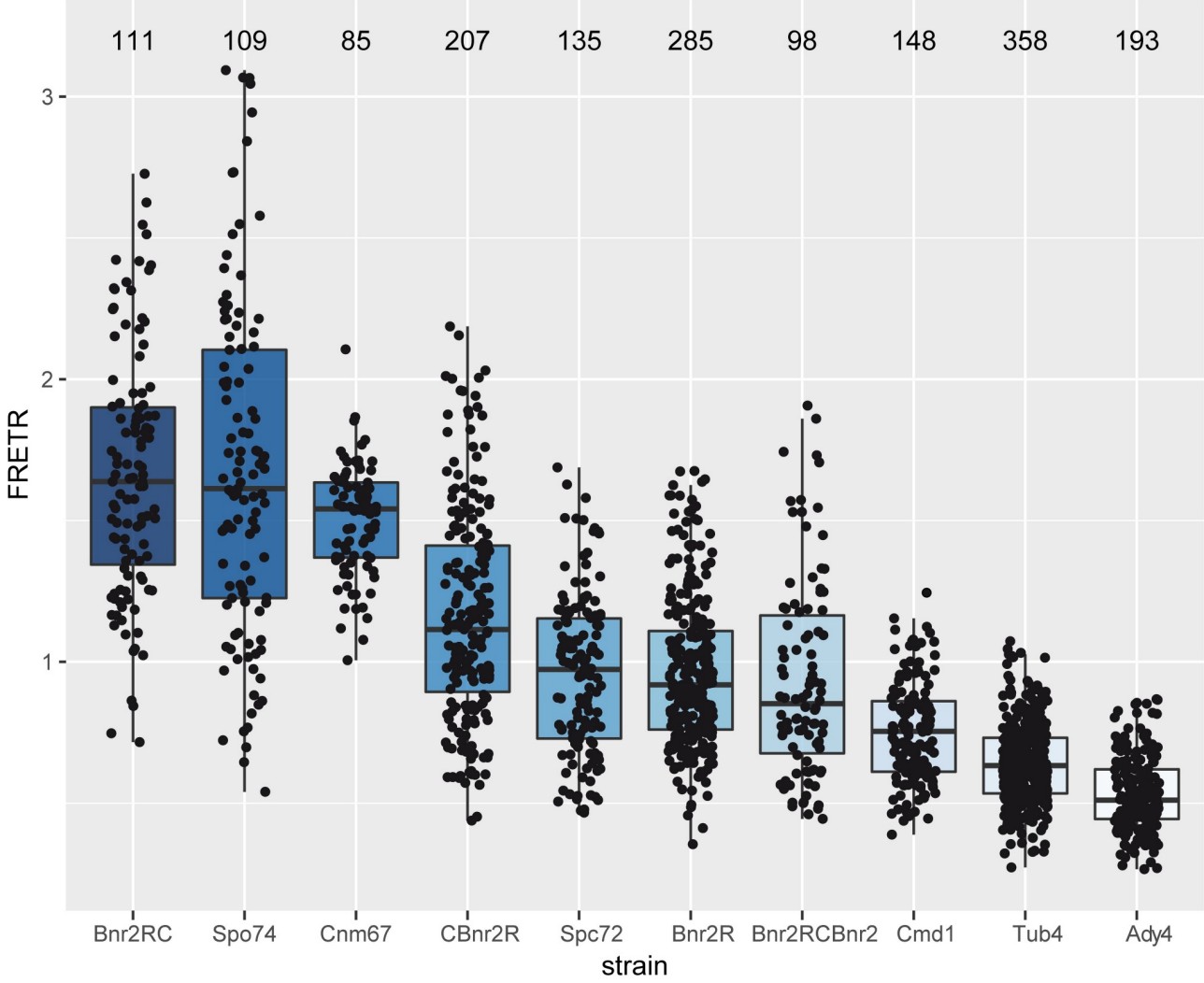

**Fig 2. Boxplot of FRETR-values measured for the indicated strain.** The FRETR-values were calculated as described in Materials and Methods. The horizontal bar in the box represents the median. The hinges represent the 25th and 75th percentiles. The whiskers extend 1.5 times the inter-quartile range from each hinge. Individual data is plotted as an overlaid scatter plot. The number plotted above each bar represents the number of individual SPBs measured for each strain.

is still around 1, suggesting no FRET, there is also a substantial amount of single measurements with FRETR values at, or even above 1.5, suggesting interaction of at least a few molecules of the tested pair of proteins. The highest FRETR values are found for AgCnm67 and AgSpo74, which produced a median FRETR slightly below our positive control, an AgBnr2 protein with yoClover and yomRuby2 fused in tandem to the carboxyterminal end of the protein.

The FRET-values obtained with the different AgBnr2 constructs do not allow a final conclusion if current models of formin regulation [30] can be applied to AgBnr2. Fig 3 shows schematic drawings of all AgBnr2 fusions together with their proposed inactive and active conformations and potentially resulting FRET. Apart from the positive control, where yomRuby2 is directly fused to yoClover, only the inactive form of CBnr2R should allow for a significant FRETR value. Even if some measurements result in a FRETR above 1.5, the majority is much lower, resulting in a median that suggests no interaction for the majority of molecules.

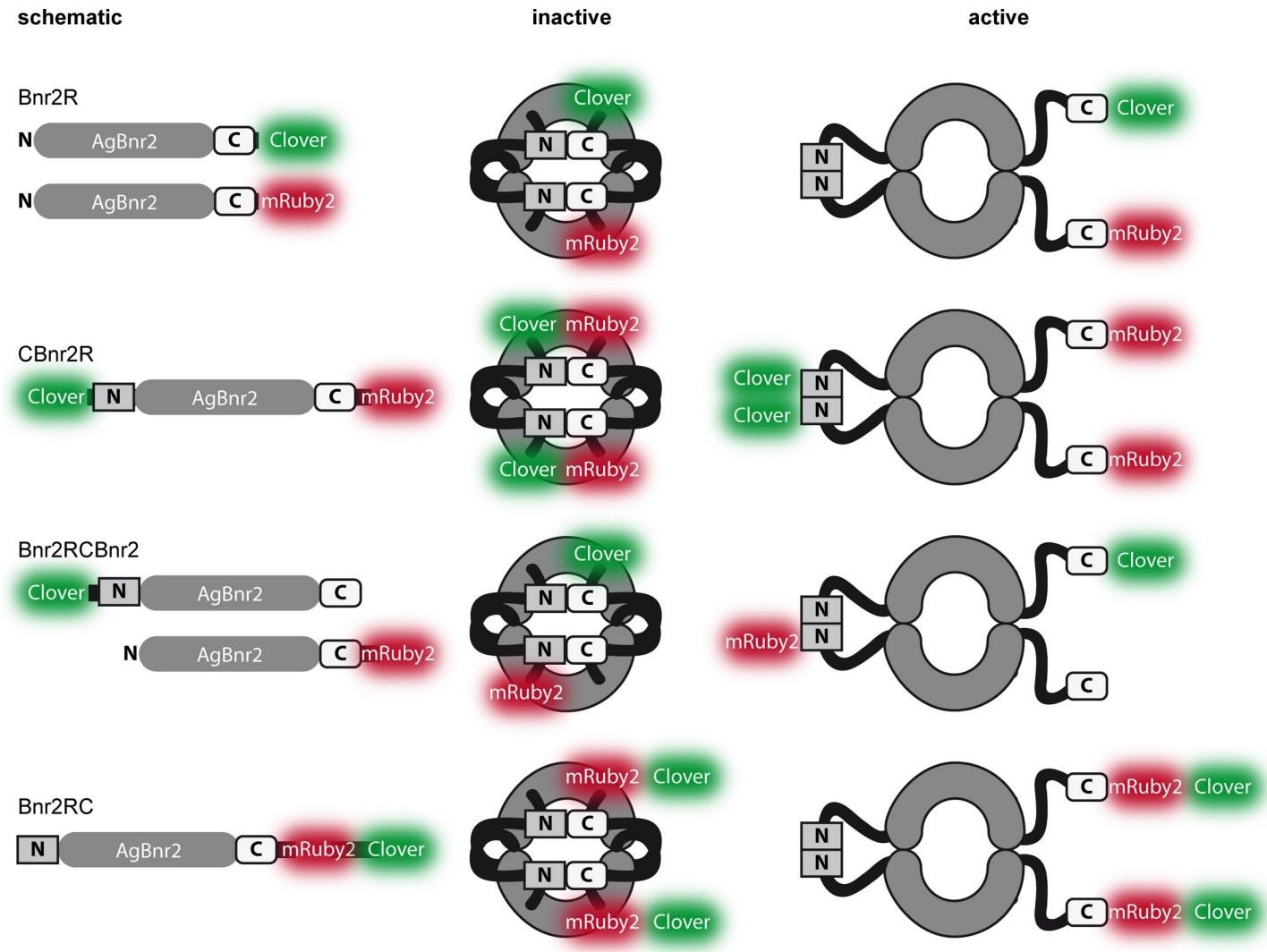

**Fig 3. Schematic and possible conformations of the different AgBnr2 constructs used for FRET-measurements.** The inactive and active forms are based on current hypotheses of formin regulation (Chesarone et al., 2010).

The high FRETR we obtained for AgSpo74 indicates that AgSpo74 may play an important role in targeting AgBnr2 to the SPB, and thus also in forming the spores. To test this hypothesis we quantified the number of SPBs with AgBnr2 signal in wild type and *Agspo74* deletion mutants. Fig 4 shows the statistics (Fig 4A) together with some sample images (Fig 4B) and the growth speed of the strains (Fig 4C). While in the wild-type strain close to 15% of the nuclei display an SPB labeled by AgBnr2-yomRuby2, below 3% of the *Δspo74* strain display such a signal. Both strains grew similar to control strains without the labeled proteins during the first 90 hours of growth (Fig 4C). The strain with both labels, but with the *SPO74* gene (left graph) grew even faster in the later growth phase. The reason for this behavior remains unclear.

In order to gain further insight into the importance of *AgSPO74* for spore formation, we further investigated the mutant phenotype in comparison to mutants of the other SOP-components AgAdy4 and AgMpc54. Deletion Mutants of all three SOP-components did not show any defects in vegetative growth. The growth speed of all three deletions was identical to the

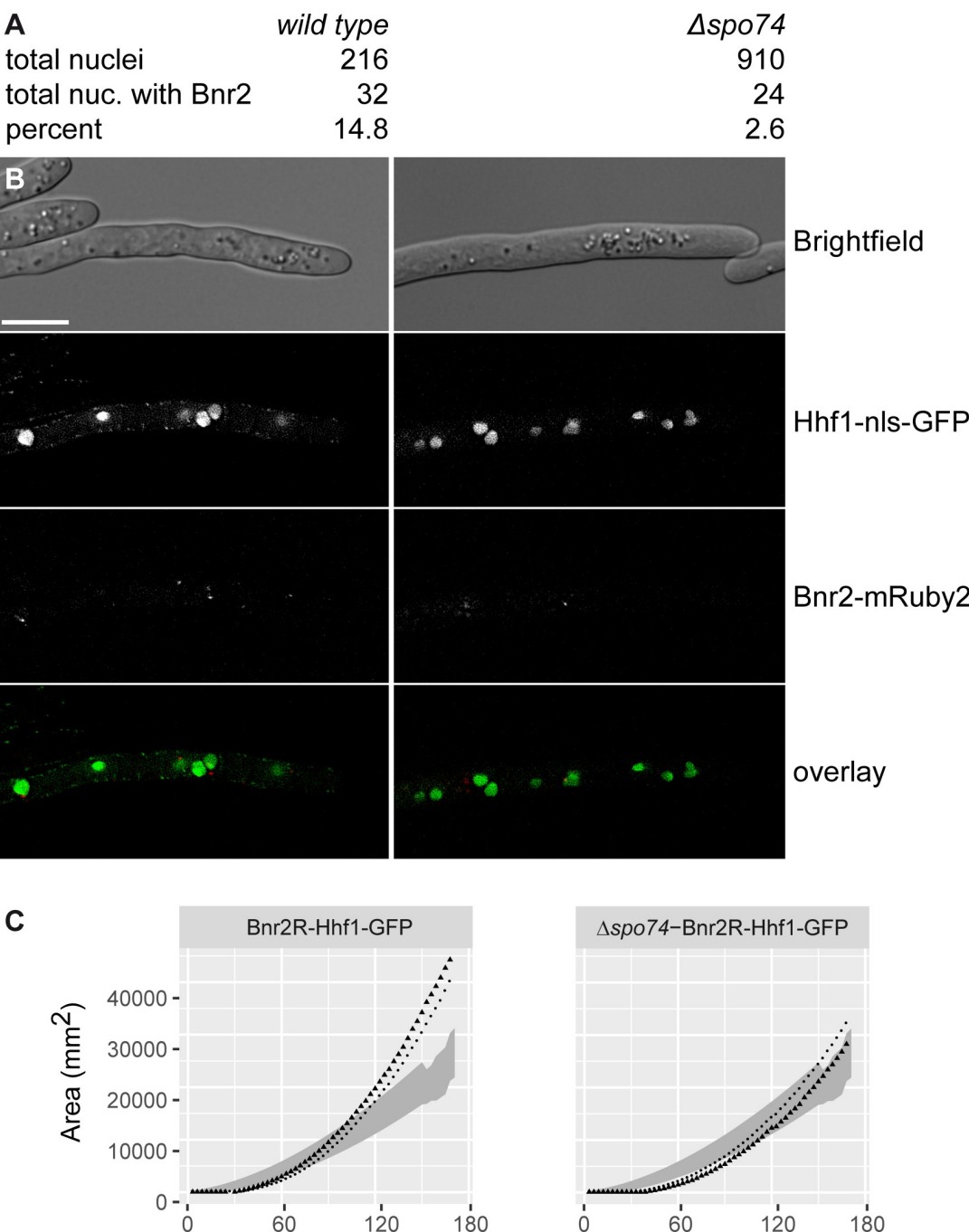

**Fig 4. Quantification of AgBnr2 signals at nuclei of wild type and deletion mutants of *Agspo74*.** A) Numbers for total nuclei and total nuclei showing AgBnr2 signals together with the percentage of nuclei showing AgBnr2 signals. B) Representative sample image from the dataset. GFP fused to the nuclear localization signal of histone h4 (HHF1) was used to mark the nuclei. The scale bar represents 10 μm. C) Radial growth of the strains from above. The graph represents the area of the mycelium growing (in mm²) measured every 3 hours over the indicated time frame.

wild-type-strain (Fig 5A and 5B) and no further defects in organization of the cell wall or the cytoskeleton were found.

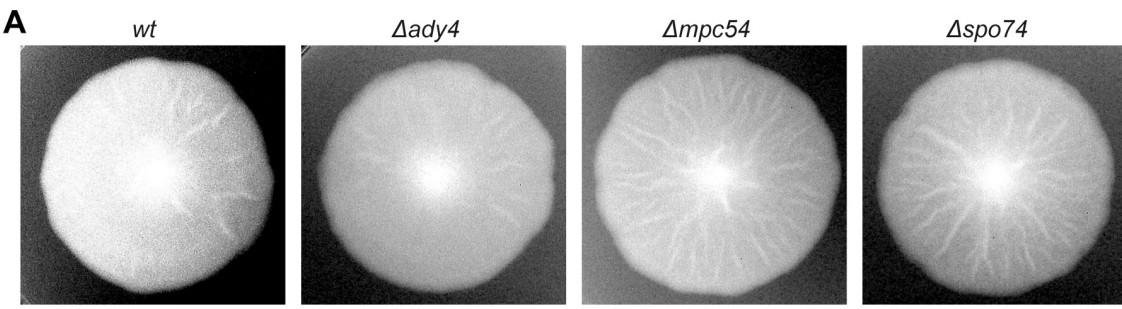

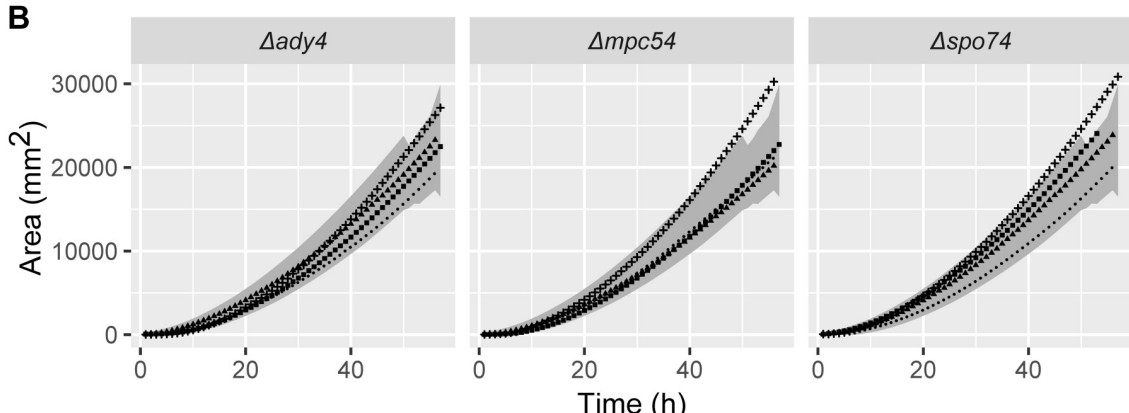

**Fig 5. Radial growth of strains carrying deletions of SOP-components.** A) Sample images of mycelia grown for 4 days. B) The graph represents the area of the mycelium growing (in mm²) measured every 3 hours over the indicated time frame.

When looking at sporulation, the results were different (Fig 6). Deletions of *Agspo74* did not sporulate at all, emphasizing the important role of the protein for spore formation. An *Agmpc54* deletion produced spores, but these spores were predominantly abnormal and approximately one third larger in size than spores from the other strains. In addition, staining of the different spore structures revealed abnormal distribution of actin, chitin, chitosan and membranes (Fig 6C). The *Agady4*-deletion showed spores similar to those of the wild-type strain, however abnormal chitin distribution was observed.

## Discussion

The results obtained in this study regarding the interaction of AgBnr2 with components of the SOP in *Ashbya gossypii* correlate well with the current model of fungal spore formation as primarily suggested by research on *Saccharomyces cerevisiae*. There, the meiotic outer plaque is assembled at the SPB as a layer sitting above the inner layer 1 of the SPB [1, 2]. The MOP is formed mainly by Spo21 and Mpc54, probably arranged with the amino-termini towards the cytoplasm and linked to the central part of the SPB via interaction of the carboxy-termini with Cnm67 [3, 4, 31]. In contrast, the position of Spo74 within the SPB is not as precisely characterized, but electron microscopy and interaction data suggest that it is an integral component of the MOP [5]. While it is already known that the general structure of the layers of the SPB of *Ashbya gossypii* seems to be conserved with slightly different dimensions [32], the data from our intensity measurements might also allow a comparison of the relative amounts of the SPB-components between *S. cerevisiae* and *A. gossypii*, at least for the proteins that were expressed

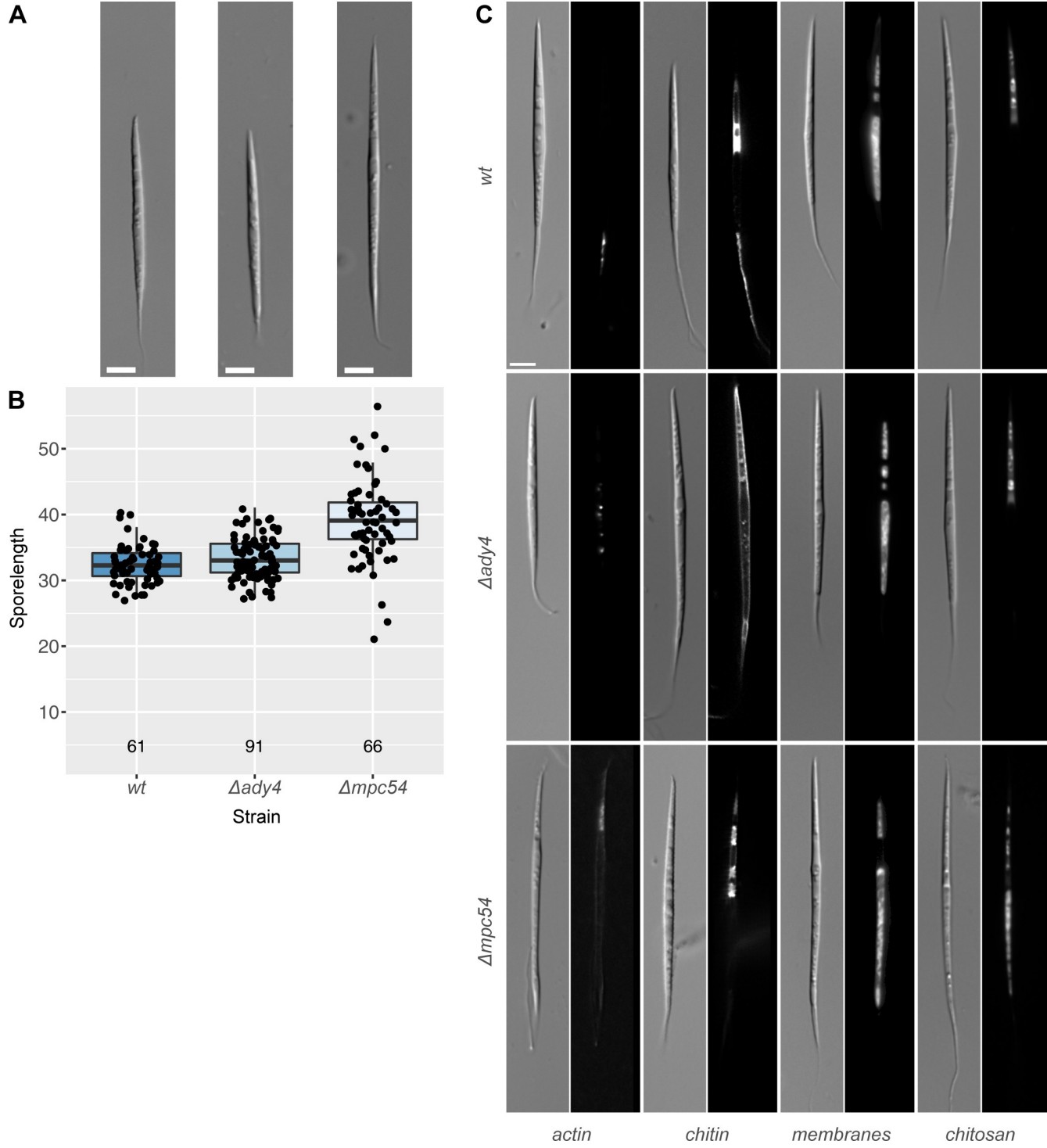

**Fig 6. Phenotypes of spores from *ady4* and *mpc54* deletion mutants.** A) Representative images of spores. The scale bar represents 5 μm. B) Boxplot of spores length measured for the indicated strain. The horizontal bar in the box represents the median. The hinges represent the 25th and 75th percentiles. The whiskers extend 1.5 times the inter-quartile range from each hinge. Individual data is plotted as an overlaid scatter plot. The number plotted below each bar represents the number of spores measured for each strain. C) Actin staining was performed using rhodamine-phalloidin; chitin was stained with calcofluor white; membranes were stained using DiOC6(3), chitosan was visualized using eosin Y.

from their own promotor. Numbers for comparable intensity measurements from *S. cerevisiae* are available for Cnm67 and Cmd1 with a ratio of about 1:1.32 [28]. For AgCnm67 and AgCmd1 we measure a ration of 1:1.95, which is in similar order of magnitude considering the high variability in our measurements.

Based on the similarity to *S. cerevisiae* we can integrate our findings into a model that allows us to speculate about a the localization and function of AgBnr2 within the SOP. Fig 7A shows a nucleus in a hypha with AgBnr2 during the early and middle phase of sporulation, finally resulting in the mature spore shown in Fig 7B. The spindle pole model shown in Fig 7C places AgBnr2 directly into the SOP-structure. However, from our current data the orientation of the AgBnr2 proteins remains unclear: We measured only minor to no FRET for AgSpc72, for which we had found a strong 2-hybrid interaction in previous work [11]. A possible reason for this might be the fact that a 2-hybrid interaction do not necessarily reflect a direct interaction. It is possible that another protein, which is present in the yeast nucleus bridges the interaction between the two proteins, which is not unlikely considering the high similarity between *A. gossypii* and *S. cerevisiae*. In contrast to AgSpc72, we were able to verify the 2-hybrid interaction of AgCnm67 with AgBnr2 from the same study [11] by our current FRET-measurements. In terms of the orientation of the protein, the latter seems to be contradictory on a first glance, because we measured FRET with a carboxyterminal fusion of the AgCnm67, while the 2-hybrid interaction was found with a 270 amino acids long fragment as well as with the 1251 amino acids long, full-length protein. But formin proteins are much more flexible concerning the orientation of amino- and carboxy-termini towards each other than the other proteins that make up the structure of the SPB. The fact that we did not find significant FRET for the series of different AgBnr2-constructs we used to test homodimer formation (Figs 2 and 3), does not contradict that AgBnr2 follow the principle shown for other formin proteins [30] with auto-regulatory binding of amino- and carboxy-termini in an inactive homodimer and side-by-side alignment of two proteins in the active form, because the only conformation shown in Fig 3 potentially leading to FRET, would be an inactive AgBnr2. Taking into account that we have previously shown interaction of an amino-terminal AgBnr2-fragment with itself and the full-length protein leaves the possibility that different conformations or activation states of the protein might allow interaction of amino- or carboxy-terminal parts with AgCnm67, depending on the activation state of the protein. The strongest FRET of our samples was measured between AgSpo74 and AgBnr2, indicating that the carboxy-termini of these two proteins come very close to each other. In addition, AgBnr2 localization to the SPB depends to large parts on AgSpo74, which further supports the FRET-data. Interestingly, the deletion of another SOP-component, AgSpo21, did not influence localization of AgBnr2 to the SPB in previous experiments [11], allowing two main conclusions: First, AgSpo74 is the main contact point for AgBnr2 at the SPB and second, AgSpo74 does not require AgSpo21 to target AgBnr2 to the SPB, because even without AgSpo21 AgBnr2 is found at the SPB.

The interaction found with AgSpo74 raises the question what the role of AgBnr2 at the SOP could be. An active AgBnr2 could assist the construction of actin-cables at the SPB [11] which serve as a route for secretory vesicles assembling the prospore membrane. Whereas the participation of secretory vesicles would be similar to sporulation in *S. cerevisiae* where multiple proteins of the secretory machinery are also involved in sporulation [4, 7, 8], actin does not seem to play a role in this yeast [33]. The functional difference in these two related organisms can be explained by the considerable difference of the spores. The small and simple round spores of *S. cerevisiae* require neither transport over larger distances nor do they require formation of more complex structures. In contrast spores from *A. gossypii* are up to 30 μm in length and display an elongated needle shape that needs to be established during sporulation [11, 12]. Such a role for actin in shaping spores would not be uncommon for fungi and is also found in the

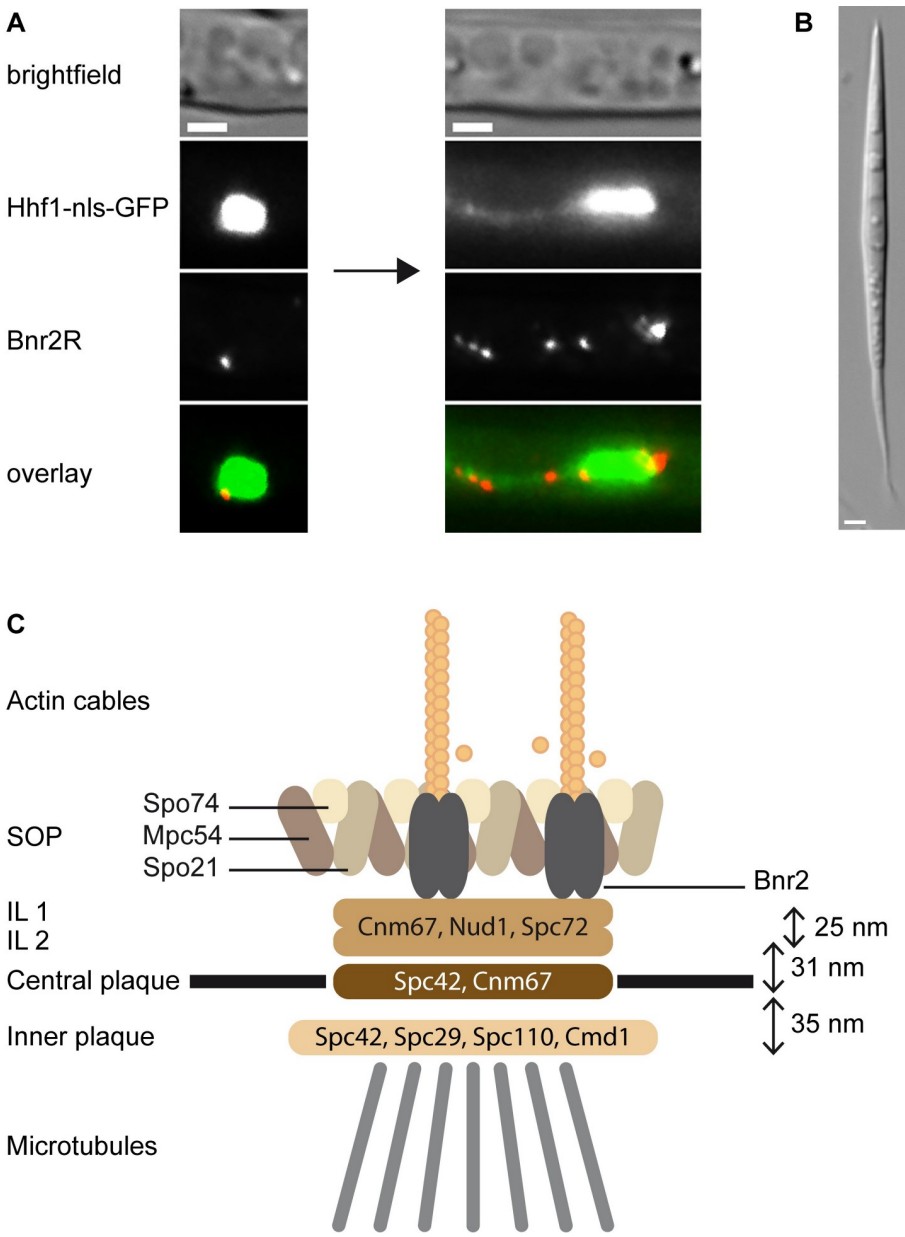

**Fig 7.** A) AgBnr2 at the nucleus during sporulation. Left: early phase, Right: middle phase of sporulation. B) Mature spore. Note that in mature spores no more fluorescent signal is detectable. C) A schematic model of an *A. gossypii* SPB during sporulation. Structural information for inner layers 1 and 2, central plaque and inner plaque is based on Lang et al. [32]. SOP and AgBnr2 arrangements were created analogous to *S. cerevisiae* combined with our data for AgBnr2.

fission yeast *Schizosaccharomyces pombe*, even though in this organism spores are not complex and of a round shape [34–36]. In this yeast meiotic actin rings are essential for proper formation of the forespore membrane and thus essential for shaping the spores.

Interestingly, the phenotype with the elongated spores, we observed for the deletion of the SOP-component *AgMPC54*, mimics the phenotype we previously found for mutants of genes of other actin regulatory proteins, namely the Rho-type GTPases AgRho1a and AgRho1b and the paxillin-homolog AgPxl1 in combination with a dominant active variant of another formin protein, AgBni1 [12]. The spores of all these mutant strains are longer than the spores from a

wild-type mycelium and like in the deletion of *AgPXL1* in combination with a dominant active AgBni1, chitosan is not only found in the anterior part of the spores, but also in the posterior part. In contrast to spores derived from the other mutants, deletion mutants of *AgMPC54* also display a defect in actin distribution within the spores: actin, which is usually only found in the posterior of spores is visible in the anterior part, if *AgMPC54* is deleted.

These additional players in sporulation compared to *S. cerevisiae* might also explain, why in contrast to *S cerevisiae*, *mpc54* deletions in *A. gossypii* still can produce spores. This is further supported by the fact that the milk yeast *Kluyveromyces lactis*, which is evolutionary closer related to *A. gossypii* then *S. cerevisiae* seems to have completely lost the *MPC54* gene during evolution (according to data in the yeast gene order browser; [37]). Nevertheless *K. lactis* has kept the ability to produce spores. This also suggests that in the lineage of the latter two fungi differences in the sporulation process of *S. cerevisiae* exist, which allow sporulation to occur without *MPC54*.

## Supporting information

**S1 Fig. Radial growth of strains carrying a single labeled SPB/SOP-component.**
(PDF)

**S2 Fig. Radial growth of strains carrying two labeled SPB/SOP-components.**
(PDF)

## Acknowledgments

This work was supported by grant SCHM 2388/2-1 from the Deutsche Forschungsge-meinschaft (DFG) and the Open Access Publishing Fund of Osnabrück University. We thank Sandra Bartels and Bernadette Sander-Turgut for excellent technical support, Lucille Schmied-ing for help with editing the manuscript and Jürgen J. Heinisch for constant support.

## Author Contributions

**Conceptualization:** Doris Nordmann, Hans-Peter Schmitz.

**Data curation:** Dario Wabner, Tom Overhageböck, Hans-Peter Schmitz.

**Formal analysis:** Dario Wabner, Tom Overhageböck, Hans-Peter Schmitz.

**Funding acquisition:** Hans-Peter Schmitz.

**Investigation:** Dario Wabner, Tom Overhageböck, Doris Nordmann, Julia Kronenberg, Florian Kramer, Hans-Peter Schmitz.

**Methodology:** Dario Wabner, Tom Overhageböck, Doris Nordmann, Julia Kronenberg, Hans-Peter Schmitz.

**Project administration:** Hans-Peter Schmitz.

**Resources:** Hans-Peter Schmitz.

**Software:** Hans-Peter Schmitz.

**Supervision:** Doris Nordmann, Hans-Peter Schmitz.

**Validation:** Doris Nordmann, Hans-Peter Schmitz.

**Visualization:** Dario Wabner, Tom Overhageböck, Doris Nordmann, Julia Kronenberg, Florian Kramer, Hans-Peter Schmitz.

**Writing – original draft:** Hans-Peter Schmitz.

**Writing – review & editing:** Dario Wabner, Tom Overhageböck, Doris Nordmann, Julia Kronenberg, Florian Kramer, Hans-Peter Schmitz.

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
