## [Decision Letter · Decision Letter 0]

19 Jul 2019

PONE-D-19-16643

Analysis of the protein composition of the spindle pole body during sporulation in Ashbya gossypii

PLOS ONE

Dear PD Dr. Schmitz,

Thank you for submitting your manuscript to PLOS ONE. After careful consideration, we feel that it has merit but does not fully meet PLOS ONE’s publication criteria as it currently stands. Therefore, we invite you to submit a revised version of the manuscript that addresses the points raised during the review process.

I will invite you to answer all the comments raised by both reviewers

You will need to specifically clarify all the major points raised by the reviewer 1 

- regarding the FRET measurements : GFP and RFP, CFP and YFP.

- the self interaction of inactive Bnr2.

- regarding the gowth of strains. Data must be presented identically to allow comparison.

- answer the questions regarding line 321-323 and line 325-327

You will also have to give provide more details (for non-experts) on the defects seen in the Agmpc54∆ and Agady4∆ mutants (see reviewer 2)

Also, please complete the Method section as requested by both reviewers.

We would appreciate receiving your revised manuscript by Sep 02 2019 11:59PM. To enhance the reproducibility of your results, we recommend that if applicable you deposit your laboratory protocols in protocols.io, where a protocol can be assigned its own identifier (DOI) such that it can be cited independently in the future. For instructions see: http://journals.plos.org/plosone/s/submission-guidelines#loc-laboratory-protocols

We look forward to receiving your revised manuscript.

Kind regards,

Claude Prigent

Academic Editor

PLOS ONE

Journal Requirements:

1. Please include captions for your Supporting Information files at the end of your manuscript, and update any in-text citations to match accordingly. Please see our Supporting Information guidelines for more information: http://journals.plos.org/plosone/s/supporting-information.

Reviewers' comments:

Reviewer's Responses to Questions

**Comments to the Author**

1. Is the manuscript technically sound, and do the data support the conclusions?

Reviewer #1: Partly

Reviewer #2: Yes

2. Has the statistical analysis been performed appropriately and rigorously? 

Reviewer #1: Yes

Reviewer #2: Yes

3. Have the authors made all data underlying the findings in their manuscript fully available?

Reviewer #1: Yes

Reviewer #2: Yes

4. Is the manuscript presented in an intelligible fashion and written in standard English?

Reviewer #1: Yes

Reviewer #2: Yes

5. Review Comments to the Author

Reviewer #1: The manuscript of Wabner et al “Analysis of the protein composition of the spindle pole body (SPB) during sporulation in Ashbya gossypii” deals with the formation of spores in Ashbya gossioii and the importance of certain spindle pole proteins during sporulation. Previous work showed that the formin Bnr2 is important for spore formation, is located at the spindle pole body and interacts with some of the spindle pole body components. In this follow up investigation, the protein-protein interactions at the spindle pole body were investigated by fluorescence resonance energy transfer (FRET). Furthermore, sporulation tests were done with some of the SPB components. The manuscript touches a very interesting topic, organization of the microtubule-organizing center during growth and sporulation. Especially, the changes associated with the different phases at the SPB, going from microtubule-organizing to actin-filament organizing deserve thorough investigation. Unfortunately, the manuscript in its current form contains some inconsistencies concerning data and the text, therefore it is premature to publish it.

My main points are:

-line 232-237: in Kemper et al. the FRET is described between GFP and RFP, CFP and YFP are mentioned only in the methods section. This is rather confusing and does not help very much in the discussion of why there are differences between the FRET measurements. I suggest the authors repeat the measurements with the original strains and their current analysis setup and include the data for comparison here, also because in the former publication no images of the FRET analysis were shown.

-line 240-250: initially, CBnr2R was categorized in the non-FRET group with frequently FRET positive signals. In the last sentence of the section, a positive FRET is claimed for the inactive form of CBnr2R, exactly as observed? Due to the high signal variation, this seems to be an over-interpretation of the data. Clearly more work is necessary to show clearly that Bnr2 shows self-interactions in its inactive form.

-line 261-263: very confusing statements: both grow similar or one faster than wt? in the Figure, the left strain seems to grow faster, however this one contains Spo74. Also figure 4b shows images and 4c the growth curves.

-line 321-323: if the authors do not fully trust the Y2H measurements of Kemper et al, why are these same Y2H interactions the main reason for the placement of the N-terminus of Bnr2 towards the IL1 in the model of figure 7? Maybe some more FRET measurements with N-terminal fusions are necessary to clarify this point and to have more support for the model.

-line 325-327: in the citation Kemper et al., I could only find the usage of RFP and GFP for FRET measurements and only the interaction between Bnr2 and Cnm67 was tested by FRET. FRET data of Bnr2 with spo21 I could not find. Moreover, the FRET data of Kemper et al between Cnm67 and the C-terminus of Bnr2 contradicts the model in figure 7, that shows a placement of the C-terminus of Bnr2 that should not result in a FRET signal together with Cnm67.

-line 364-351: too much speculation, see also the comment on Cnm67 FRET results.

Minor points:

-the FRET measurements have to be described in more detail. There are several crucial information that the reader is supposed to look after in diverse publications. These should be presented in the methods section as well as a good description of the adaptations that have been performed to the FRETSCAL algorithm. This will be important for everybody doing similar measurements in other fungi.

- line 218-222: is there an estimation of how much of a mycelium was producing spores at a later time point? This information might be helpful to interpret the FRET data.

Reviewer #2: This study from Wabner et al expands on earlier studies from the Schmitz group demonstrating the importance of the actin cytoskeleton for formation of the complex needle-shaped spores in A. gossypii and showing that the actin-nucleating formin AgBnr2 localizes to the spindle pole body during sporulation. In this study, FRET interactions between AgBnr2 and different SPB proteins are used to show that AgBnr2 is located close to the sporulation outer plaque (SOP) protein AgSpo74. Consistent with this, deletion of AgSPO74 greatly reduces recruitment of AgBnr2 to the SOP. The presence of AgBnr2 in the SOP is in contrast to the homologous structure in S. cerevisiae, the meiotic outer plaque (MOP). Strengthening the contrast with S. cerevisiae, the authors delete the Ashbya orthologs of three MOP components important for sporulation in S. cerevisiae find that while AgSPO74 is essential for spore formation, AgMPC54 an AgADY4 are not.

Specific comments:

For the non-expert it would be helpful to more fully explain the defects in actin and/or chitin distribution seen in the spores of the Agmpc54∆ and Agady4∆ mutants. As only one cell is shown, it is not so clear how the mutants are abnormal. For Agmpc54∆, the authors state that similar defects were seen in mutants of rho1 and other actin-associated genes. I assume this refers to the elongated spore phenotype and not necessarily the effects on the distribution of actin, chitin, etc.? It appears in the earlier paper that not all of the mutations that caused lengthening of the spores produced the same effects on chitin distribution. Does Agmpc54∆ resemble a specific mutant? This discussion should be clarified.

I may be overlooking it, but I do not see in the Methods a description of how the CFW/DiOC6/Eosin Y stainings were performed.

As part of their data collection for the FRET measurements, the authors should have generated measurements of the fluorescence intensity of mRuby fusions to multiple different SPB components. These values could be used to determine the relative stoichiometries of the different components. It would be of interest to compare these values to stoichiometries reported for the S. cerevisiae SPB.

Based on the studies in S. cerevisiae, where loss of either SPO74 or SPO21 causes complete absence of the MOP, it is surprising that AgSPO74 is important for AgBnr2 recruitment to the SOP but that AgSPO21 is not (the latter result is reported in an earlier paper from the Schmitz lab). It would be nice to revisit this second result here, but at a minimum this difference should be discussed.

6. PLOS authors have the option to publish the peer review history of their article (what does this mean?). If published, this will include your full peer review and any attached files.

Reviewer #1: Yes: Christof Taxis

Reviewer #2: No

---

## [Author Response · Author response to Decision Letter 0]

29 Aug 2019

Reviewer comment:

-line 232-237: in Kemper et al. the FRET is described between GFP and RFP, CFP and YFP are mentioned only in the methods section. This is rather confusing and does not help very much in the discussion of why there are differences between the FRET measurements. I suggest the authors repeat the measurements with the original strains and their current analysis setup and include the data for comparison here, also because in the former publication no images of the FRET analysis were shown.

Answer:

The statement of the reviewer is correct that there is a confusing, false statement in Kemper et al. (2011). The fusions used for the experiments in this study were yEGFP fused to AgBnr2 and tdtomato fused to Cnm67. We included measurements with these original strains in the data and also exchanged the images of the AgCnm67 fusion to the tdtomato version.

Reviewer comment:

-line 240-250: initially, CBnr2R was categorized in the non-FRET group with frequently FRET positive signals. In the last sentence of the section, a positive FRET is claimed for the inactive form of CBnr2R, exactly as observed? Due to the high signal variation, this seems to be an over-interpretation of the data. Clearly more work is necessary to show clearly that Bnr2 shows self-interactions in its inactive form.

Answer:

We have changed the statement in the text and rewrote the paragraph in the discussion about AgBnr2 self interaction.

Reviewer comment:

-line 261-263: very confusing statements: both grow similar or one faster than wt? in the Figure, the left strain seems to grow faster, however this one contains Spo74. Also figure 4b shows images and 4c the growth curves.

Answer:

The confusion here is caused because the text refers to an earlier version of the figure, which we have changed afterward. We have to apologize for overlooking that. We changed the text to “Both strains grew similar to control strains without the labeled proteins (Figure 4C) during the first 90 hours of growth. The strain with both labels, but with the SPO74 gene (left graph) grew even faster in the later growth phase.”

Reviewer comment:

-line 321-323: if the authors do not fully trust the Y2H measurements of Kemper et al, why are these same Y2H interactions the main reason for the placement of the N-terminus of Bnr2 towards the IL1 in the model of figure 7? Maybe some more FRET measurements with N-terminal fusions are necessary to clarify this point and to have more support for the model.

Answer:

We including the measurements with AgCnm67 fused to a different fluorophore, which now yield significant FRETR values. This resolves the discrepancy between the old FRET-data that did not display interaction and the 2-hybrid data. We have changed the text accordingly.

Reviewer comment:

-line 325-327: in the citation Kemper et al., I could only find the usage of RFP and GFP for FRET measurements and only the interaction between Bnr2 and Cnm67 was tested by FRET. FRET data of Bnr2 with spo21 I could not find. Moreover, the FRET data of Kemper et al between Cnm67 and the C-terminus of Bnr2 contradicts the model in figure 7, that shows a placement of the C-terminus of Bnr2 that should not result in a FRET signal together with Cnm67.

Answer:

We have changed the model to integrate the novel measurements of AgCnm67 and AgBnr2. From our data a conclusion about the orientation of AgBnr2 within the SPB is not possible, in addition it is, in contrast to the main SPC components also probably not static, but dynamic. We changed the discussion accordingly. The reviewer is also correct with AgSpo21. Kemper et al. contains colocalization-data, but no FRET measurements. This was confused with the AgSPC42-FRET measurements in the same paper. We removed all wrong references from the manuscript.

Reviewer comment:

-line 364-351: too much speculation, see also the comment on Cnm67 FRET results.

Minor points:

Answer:

We removed the parts containing speculation from the discussion and included discussion of the novel Cnm67 measurements.

Reviewer comment:

-the FRET measurements have to be described in more detail. There are several crucial information that the reader is supposed to look after in diverse publications. These should be presented in the methods section as well as a good description of the adaptations that have been performed to the FRETSCAL algorithm. This will be important for everybody doing similar measurements in other fungi.

Answer:

We have included a more detailed description of the FRET-measurements and a more detailed description of the analysis procedure.

Reviewer comment:

- line 218-222: is there an estimation of how much of a mycelium was producing spores at a later time point? This information might be helpful to interpret the FRET data.

Answer:

This is unfortunately a major problem in our system, which, so far we could not resolve. The fact that we might have looked at non sporulating SPBs in many cases is probably also responsible for the high variability in our measurements. We have undertaken many attempts to quantify sporulation reliably. Even though, some methods have been used before by other groups, none of these methods yields data that results in comparable numbers, even if the same culture is quantified multiple times, in our hands. In addition, there is another difficulty with quantifying sporulation under the microscope. During our measurements we realized that light seems to have a negative effect on sporulation. We have already looked deeper into that and identified potential factors involved in that regulation, which we will further investigate.

Reviewer comment:

For the non-expert it would be helpful to more fully explain the defects in actin and/or chitin distribution seen in the spores of the Agmpc54∆ and Agady4∆ mutants. As only one cell is shown, it is not so clear how the mutants are abnormal. For Agmpc54∆, the authors state that similar defects were seen in mutants of rho1 and other actin-associated genes. I assume this refers to the elongated spore phenotype and not necessarily the effects on the distribution of actin, chitin, etc.? It appears in the earlier paper that not all of the mutations that caused lengthening of the spores produced the same effects on chitin distribution. Does Agmpc54∆ resemble a specific mutant? This discussion should be clarified.

Answer:

We described in more detail, which phenotypes are similar to which other mutant and which phenotypes differ from the ones described before.

Reviewer comment:

I may be overlooking it, but I do not see in the Methods a description of how the CFW/DiOC6/Eosin Y stainings were performed.

Answer:

This information was indeed missing. The staining is described in detail in Lickfeld et al. (2012). We have added a statement and the reference to the methods section.

Reviewer comment:

As part of their data collection for the FRET measurements, the authors should have generated measurements of the fluorescence intensity of mRuby fusions to multiple different SPB components. These values could be used to determine the relative stoichiometries of the different components. It would be of interest to compare these values to stoichiometries reported for the S. cerevisiae SPB.

Answer:

We have integrated the data into Table 2 and added a paragraph discussing differences to Saccharomyces cerevisiae. In the discussion we focused on the components that were expressed from their own protomer and for which we could find comparable data in the literature.

Reviewer comment:

Based on the studies in S. cerevisiae, where loss of either SPO74 or SPO21 causes complete absence of the MOP, it is surprising that AgSPO74 is important for AgBnr2 recruitment to the SOP but that AgSPO21 is not (the latter result is reported in an earlier paper from the Schmitz lab). It would be nice to revisit this second result here, but at a minimum this difference should be discussed.

Answer:

We added a paragraph discussing this to the manuscript.

---

## [Decision Letter · Decision Letter 1]

10 Sep 2019

[EXSCINDED]

PONE-D-19-16643R1

Analysis of the protein composition of the spindle pole body during sporulation in Ashbya gossypii

PLOS ONE

Dear PD Dr. Schmitz,

Thank you for submitting your manuscript to PLOS ONE. After careful consideration, we feel that it has merit but does not fully meet PLOS ONE’s publication criteria as it currently stands. Therefore, we invite you to submit a revised version of the manuscript that addresses the two minor points raised during the review process.

--Line 265-266: Figure 4A shows statistics, Figure 4B the images, the text is not correct here.

- Line 366: "...actin does not seem to play a role in this yeast." Please provide a reference for this statement.

We would appreciate receiving your revised manuscript by Oct 25 2019 11:59PM. To enhance the reproducibility of your results, we recommend that if applicable you deposit your laboratory protocols in protocols.io, where a protocol can be assigned its own identifier (DOI) such that it can be cited independently in the future. For instructions see: http://journals.plos.org/plosone/s/submission-guidelines#loc-laboratory-protocols

We look forward to receiving your revised manuscript.

Kind regards,

Claude Prigent

Academic Editor

PLOS ONE

Reviewers' comments:

Reviewer's Responses to Questions

**Comments to the Author**

1. If the authors have adequately addressed your comments raised in a previous round of review and you feel that this manuscript is now acceptable for publication, you may indicate that here to bypass the “Comments to the Author” section, enter your conflict of interest statement in the “Confidential to Editor” section, and submit your "Accept" recommendation.

Reviewer #1: (No Response)

Reviewer #2: All comments have been addressed

2. Is the manuscript technically sound, and do the data support the conclusions?

Reviewer #1: Yes

Reviewer #2: Yes

3. Has the statistical analysis been performed appropriately and rigorously? 

Reviewer #1: Yes

Reviewer #2: Yes

4. Have the authors made all data underlying the findings in their manuscript fully available?

Reviewer #1: Yes

Reviewer #2: Yes

5. Is the manuscript presented in an intelligible fashion and written in standard English?

Reviewer #1: Yes

Reviewer #2: Yes

6. Review Comments to the Author

Reviewer #1: My comments to the manuscript have been almost addressed, only two minor things remain:

-Line 265-266: Figure 4A shows statistics, Figure 4B the images, the text is not correct here.

line 366: "...actin does not seem to play a tole in theis yeast." Please provide a reference for this statement.

Reviewer #2: (No Response)

7. PLOS authors have the option to publish the peer review history of their article (what does this mean?). If published, this will include your full peer review and any attached files.

Reviewer #1: Yes: Christof Taxis

Reviewer #2: No

---

## [Author Response · Author response to Decision Letter 1]

13 Sep 2019

Reviewer comment:

-Line 265-266: Figure 4A shows statistics, Figure 4B the images, the text is not correct here.

We have changed the text to:

“Figure 4 shows the statistics (Figure 4A) together with some sample images (Figure 4B) and the growth speed of the strains (Figure 4C).”

Which now fits to the labeling in the figure.

Reviewer comment:

- Line 366: "...actin does not seem to play a role in this yeast." Please provide a reference for this statement.

We have added the reference:

“Taxis C, Maeder C, Reber S, Rathfelder N, Miura K, Greger K, Stelzer EH, Knop M. Dynamic organization of the actin cytoskeleton during meiosis and spore formation in budding yeast. Traffic. 2006;7(12):1628-42. PubMed PMID: 17118118.”

---

## [Editor Report · Decision Letter 2]

20 Sep 2019

Analysis of the protein composition of the spindle pole body during sporulation in Ashbya gossypii

PONE-D-19-16643R2

Dear Dr. Schmitz,

We are pleased to inform you that your manuscript has been judged scientifically suitable for publication and will be formally accepted for publication once it complies with all outstanding technical requirements.

With kind regards,

Claude Prigent

Academic Editor

PLOS ONE
---

## [Editor Report · Acceptance letter]

25 Sep 2019

PONE-D-19-16643R2 

Analysis of the protein composition of the spindle pole body during sporulation in *Ashbya gossypii*

Dear Dr. Schmitz:

I am pleased to inform you that your manuscript has been deemed suitable for publication in PLOS ONE. Congratulations! Your manuscript is now with our production department. 

With kind regards,

on behalf of

Dr. Claude Prigent 

Academic Editor

PLOS ONE